# Impact of cerebrospinal fluid leukocyte infiltration and activated neuroimmune mediators on survival with HIV-associated cryptococcal meningitis

Samuel Okurut[1,2]*, David R. Boulware[3], Yukari C. Manabe[1,4], Lillian Tugume[1], Caleb P. Skipper[3], Kenneth Ssebambulidde[1], Joshua Rhein[3], Abdu K. Musubire[1], Andrew Akampurira[1,2], Elizabeth C. Okafor[3], Joseph O. Olobo[5], Edward N. Janoff[6,7], David B. Meya[1,3,8], for ASTRO Trial Team

1 Research Department, Infectious Diseases Institute, Makerere University, Kampala, Uganda, 2 Department of Medical Microbiology, School of Biomedical Sciences, College of Health Sciences, Makerere University, Kampala, Uganda, 3 Division of Infectious Diseases and International Medicine, Department of Medicine, University of Minnesota, Minneapolis, Minnesota, United States of America, 4 Division of Infectious Diseases, Department of Medicine, John Hopkins University School of Medicine, Baltimore, Maryland, United States of America, 5 Department of Immunology and Molecular Biology, School of Biomedical Sciences, College of Health Sciences, Makerere University, Kampala, Uganda, 6 Mucosal and Vaccine Research Program Colorado, Department of Medicine, Division of Infectious Diseases, University of Colorado Denver, Aurora, Colorado, United States of America, 7 Rocky Mountain Regional Veterans Affairs Medical Center, Aurora, Colorado United States of America, 8 Department of Medicine, School of Medicine, College of Health Sciences, Makerere University, Kampala, Uganda

* okuruts@gmail.com

**Data Availability Statement:** The article contains the dataset that informed the manuscript. The additional information that might be necessary to

## Abstract

### Introduction

Cryptococcal meningitis remains a prominent cause of death in persons with advanced HIV disease. CSF leukocyte infiltration predicts survival at 18 weeks; however, how CSF immune response relates to CSF leukocyte infiltration is unknown.

### Methods

We enrolled 401 adults with HIV-associated cryptococcal meningitis in Uganda who received amphotericin and fluconazole induction therapy. We assessed the association of CSF leukocytes, chemokine, and cytokine responses with 18-week survival.

### Results

Participants with CSF leukocytes ≥50/microliter had a higher probability of 18-week survival compared with those with ≤50 cells/microliter (68% (52/77 vs. 52% (151/292); Hazard Ratio = 1.63, 95% confidence interval 1.14–2.23; p = 0.008). Survival was also associated with higher expression of T helper (Th)-1, Th17 cytokines, and immune regulatory elements. CSF levels of Programmed Death-1 Ligand, CXCL10, and Interleukin (IL)-2 independently predicted survival. In multivariate analysis, CSF leukocytes were inversely associated with CSF fungal burden and positively associated with CSF protein and immune parameters

replicate the manuscript is either cited on in text references or provided in the cited supporting materials.

**Funding:** This work was funded in part by the National Institute of Allergy and Infectious Diseases grant R01AI078934, R01AI108479, R01AI162786, T32AI055433 to DBM and K24AI184270 to DRB and R01AI108479 to ENJ. National Institute of Neurologic Diseases and Stroke grant R01NS086312, R25TW009345, and K01TW010268 to JR. Fogarty International Center training grant R01NS086312 to JR and D43TW009771 grant to YCM. Fogarty Northern Pacific Global Health Fellowship Program grant D43TW009345 support to LT. United Kingdom Medical Research Council, Wellcome Trust, and the Department for International Development (MRC MR/M007413/1 grant to JR, Wellcome Trust 107742/Z/15/Z grant to DBM and Wellcome Trust (Training Health Researchers into Vocational Excellence (THRiVE) in East Africa 087540 grant to DBM. THRiVE-2 DELTAS Africa Initiative DEL-15-011 grant support to DBM. GlaxoSmithKline Trust in Science Africa COL100044928 grant to SO. Grand Challenges Canada S4-0296-01 grant to DBM. Veterans Affairs Research Service I01CX001464 grants to ENJ. Funders and the funding agencies had no role in the study design; collection, analysis, and interpretation of data; writing of the paper; and/or decision to submit for publication.

**Competing interests:** SO was a Fogarty and GlaxoSmithKline-Trust in Science Africa funded doctoral scholar at Infectious Diseases Institute, Makerere University. Part of the work contributed to the doctoral thesis defended at the Makerere University and cited in this article on reference 40. AMK was a member of a study data safety monitoring board.

(interferon-gamma (IFN-γ), IL-17A, tumor necrosis factor alpha (TNF)-α, and circulating $CD4^+$ and $CD8^+$ T cells).

## Conclusion

18-week survival after diagnosis of cryptococcal meningitis was associated with higher CSF leukocytes at baseline with greater T helper 1 (IFN-γ, IL-2 and TNF-α cytokines), T helper 17 (IL-17A cytokine) and $CXCR3^+$ T cell (CXCL10 chemokine) responses. These results highlight the interdependent contribution of soluble and cellular immune responses in predicting survival and may support potential pathways for adjunctive immune therapy in HIV-associated cryptococcal meningitis.

### Author summary

Cryptococcal meningitis, an infection of the brain, spinal cord, and cerebrospinal fluid (CSF), is a major contributor to death among people with advanced HIV disease, with $CD4^+$ T cells less than 100 cells/mm³. In cryptococcal meningitis it is unknown how CSF white blood cells compare to CSF immune responses, fungal burden, and patients' survival. We determined the relationship between levels of CSF immune responses, CSF fungal burden, and CSF white blood cells at the time of diagnosis with 18-week survival. Subjects who died within 18-weeks of diagnosis had lower levels of interferon (IFN-γ), interleukin (IL-2 and IL-17A) tumor necrosis factor-alpha (TNF-α) and chemokine (CXCL10) and higher fungal burden compared with survivors. Numbers of CSF white blood cells showed a positive association with CSF immune response and patient survival. However, these observations were negatively associated with CSF fungal burden. These findings are unique and provide new insights into the immunopathology of HIV-associated cryptococcal meningitis and fatal outcomes. The results suggest immune-based approaches to improve patients' survival.

## 1. Background

Cryptococcal meningitis (CM) remains one of the leading causes of AIDS-related death worldwide [1–3]. The depletion of a protective immune response with uncontrolled HIV infection is the main factor responsible for cryptococcal infection among people with advanced HIV disease. Cryptococcal evasion of the host immune response results in dissemination to the central nervous system (CNS), activating CNS-resident and patrolling immune cells directly through antigen presenting cellular responses and/or indirect through bystander activation, to exacerbate inflammation associated with the clinical disease [4–6]. Activated leukocytes in the CNS or cerebrospinal fluid (CSF) produce chemokines, cytokines, and other immune-mediating factors responsible for shaping the course of infection, immunopathology and survival [7,8]. At diagnosis and during ensuing treatment the presence of an evoked elevated or preserved immune response is observed to potentially work in synergy with anti-fungal medications to influence lower fungal burden, possible faster fungal clearance, and improved host recovery [9–11].

Among risk factors for debilitating cryptococcosis, low numbers of leukocytes in CSF are a harbinger of HIV-related immune suppression, high fungal burden, and cryptococcosis-

related mortality with advanced HIV disease [12,13]. Monocytes, the precursors of macrophages, are important cells in which *Cryptococcus* replicates but is intracellularly shielded from immune responses. The patients with cryptococcal meningitis with lower probability of survival have impaired monocyte immune activation and altered cryptococcal phagocytic function [14,15]. Human hosts with increased cryptococcal phagocytosis by macrophages have increased intracellular macrophage fungal replication, high CSF fungal burden and lower probability of recovery [15]. Thus, the potential disconnect between phagocytosis and fungal killing may influence cryptococcal disease progression. Unchecked control of intracellular macrophage fungal replication with high fungal burden may result from low numbers of circulating $CD4^+$ T cells helper response and the paucity of soluble immune mediators that hinder fungal killing by macrophages [16].

The immune-activated cytokines, chemokines, and checkpoint regulatory responses are important host factors elevated among patients with lower CSF fungal burden. The Th1 cytokine interferon (IFN)-γ elicits signal transduction to activate intracellular pathogen killing among intracellular infected macrophages [17]. The CXCL10 or interferon gamma inducible protein 10 (IP-10) supports the recruitment of activated $CXCR3^+$ T cells, Natural Killer cells (NK cells) and NK T cells to mediate Th1-associated immune response [18]. Women with low CSF levels of the $CXCR3^+$ T cell chemoattractant chemokine CXCL10 and Th17 T cell activating cytokine IL-17A had lower probability of recovery on anti-fungal therapy [19]. These observations suggest a possible irreversible host immune and survival selection pressure that impairs host recovery, despite antifungal therapy.

Casadevall and Pirofski's immune-pathogen damage response framework theory suggests that an optimal treatment strategy for infectious diseases should enhance pathogen killing and control the potential detrimental bystander effect of the host-directed immune response mounted against the pathogen [20]. Striking a balance between immune- and antifungal-mediated cryptococcal killing mechanisms while limiting bystander neuroimmunopathology is a challenge in HIV-associated cryptococcal meningitis, especially among individuals with severe immune deficiency [21–23]. To understand immune-associated survival mechanisms with infiltrating CSF leukocytes, we examined whether the expression of CSF Th1, Th17 cytokines, and chemokine responses correlate with levels of CSF infiltrating leukocytes, CSF fungal burden, and 18-week survival.

## 2. Methods

### 2.1. Ethics Statement

The parent trial was approved by the Mulago National Referral Hospital Research and Ethics Committee (approval number: MREC—429), the University of Minnesota institutional review board (approval number: UMN IRB - 1304M31361), Uganda National Council for Science and Technology (approval number: UNCST—HS1406), the Uganda National Drug Authority (approval number: 189/ESR/NDA/DID-07/2013), and the United States Food and Drug Administration Investigational New Drug (approval number: FDA IND—120441). Participants or their surrogates provided written signed informed and storage consent for use of their specimens and data in the meningitis studies. Waiver of consent to use data and specimens in the current study was approved by the School of Biomedical Sciences, College of Health Sciences, Makerere University (approval number: SBS-REC 701).

### 2.2. Parent trial, participants, site, and setting

We included 401 of 460 consenting adults who had both CSF white cell count and CSF cytokine measurements performed at baseline among participants enrolled in the Adjunctive

Sertraline for the Treatment of Cryptococcal Meningitis trial (ASTRO Phase 3 trial) (Clinical-Trials.gov: NCT 01802385) [24,25]. Excluded were participants/samples with a variety of reasons including low sample volumes that could not support both microbiological and cytokine testing or, specimen draws that occurred on weekends/holidays where research testing was not feasible, or laboratory errors leading to missing samples or results.

Participants were recruited from Mulago and Kiruddu National Referral Hospitals in Kampala and Mbarara Regional Referral Hospital in Mbarara, Uganda between March 2015 and May 2017. Cryptococcal meningitis was diagnosed using CSF cryptococcal antigen (CrAg) lateral flow assay (Immy Inc., Norman, Oklahoma, USA) and laboratory quantitative culture CSF fungal colony forming units, (CFU). In the trial, there was no difference in survival by participants' randomization into either Sertraline (trial drug arm) or placebo (standard anti-fungal therapy) [24].

The CSF was collected via lumbar puncture. The CSF leukocytes were counted in fresh CSF using a hemocytometer. The CSF was centrifuged at 500g for 5 minutes and the supernatant separated and cryopreserved at -80˚C storage before thawing for batch testing.

## 2.3. Study design

This was an exploratory study, designed to investigate the association between the levels of existing baseline CSF leukocytes demonstrated by the CSF white cell counts and the associated immune responses at CM diagnosis and 18-week survival after diagnosis or enrolment. The 18-week survival was the trial primary endpoint. Participants were systematically selected and stratified by CSF leukocyte number: ≤50/µL, 51–200 cells/µL, and 201–500 cells/µL. The initial data stratification was based on small *a priori* ranges as illustrated in (S1 Fig). This initial experimental analysis informed down-stream grouped analysis represented in all figures and tabulated data. The rationale was explored and observed unique attributes associated with immune response, fungal burden, and host survival outcome attributes that could inform new and unique biologically plausible ranges for future CM investigations.

## 2.4. Luminex cytokine and chemokine immunophenotyping

Baseline CSF cytokine and chemokine levels were measured using 1:2 dilutions with a human XL cytokine discovery kit per the manufacturer's instructions (R&D, Minneapolis, MN). The Luminex CSF data acquisition was performed at the University of Minnesota as earlier explained [19,26]. Briefly, the Th1 cytokines regulated through T-bet and STAT1 transcription factors were TNF-α, IFN-γ, IL-2, and IL-12p70. The Th2 cytokines regulated through Gata 3 and STAT 6 transcription factor-modulated cytokines were IL-4 and IL-13. The T follicular helper adaptive cells activating cytokines regulated through Bcl-6 and STAT 3 transcription factors were IL-6 and IL-10. The Th17 cytokines include IL-17A. The innate-like cytokines were IL-15, IL-8/CXCL8 or CXCL8, IL-1RA or IL-1F3 produced by innate lymphoid and myeloid cells among neutrophils, monocytes, macrophages, dendritic cells to mediate cellular chemoattraction to neuroinflammation.

The chemokines that work in synergy with induced cytokines and among activated cells to attract B cells, T cells, and innate lymphoid cells resulting in neuroinflammation were CXCR3+ T cell activating chemokine CXCL10/IP-10 secreted by monocytes, macrophages, dendritic cells, and in the CNS secreted by microglia cells and astrocytes for lymphocyte chemoattraction to neuroinflammation. The CCL11/Eotaxin for myelocyte chemoattraction to neuroinflammation. The IL-8/CXCL8 for neutrophil activation and chemoattraction to neuroinflammation. The immune checkpoint inhibitor was PD-L1/B7-H1 for control of the resultant immune response.

## 2.5. Statistical analysis

Data were analyzed using GraphPad Prism version 9.3.0 (San Diego, California, USA). In the univariate analyses, continuous variables were analyzed using the Mann-Whitney non-parametric unpaired t-test for comparison of sample medians. The difference in survival (binary outcome) was determined using univariate Log-Rank test and multivariate Logistics Regression analysis. The 7.9% (32/401) participants with missing survival outcome data were included in the survival sensitivity analysis following a systematic statistical imputation approach to account for participants who had missing survival outcome reports during follow up. The goal was to test whether participants with missing survival outcome data reduced the power to detect the statistical differences among survival outcomes influenced these variables. The statistical analysis hypothesis tested was that all participants missing survival outcome data were either all alive or all dead approach for their data inclusion in survival sensitivity queried analysis. Hence, the results are reported as the main results without missing survival data imputation, or supplementary results with all alive or all dead missing survival data imputed findings. Missingness was reported because the trial operated in real-world setting where enrolled hospitalized participants were followed up for 18 weeks for survival outcomes outside the hospital setting. Participants were considered lost to follow up following three failed contact attempts by a phone call. Kruskal Wallis test (Analysis of variance; ANOVA) was used for simple linear group-wise analysis. Principal Component Analysis (PCA), Multivariate Linear and Logistic Regression and Person Correlation was used for complex multivariate data analysis and data stratification and among the outcome models. The p-value <0.050 at 95% confidence interval (CI) was reported as statistically significant.

## 3. Results

### 3.1. CSF leukocyte infiltration negatively correlates with CSF fungal burden and positively correlates with CSF protein and peripheral CD4$^+$ and CD8$^+$ T cell counts

Participants included 241 males and 160 females, with a median age of 35 years, (interquartile range [IQR]; 29–40 years) with confirmed HIV-associated cryptococcal meningitis. We defined the baseline participant demographics, clinical, microbiologic, and immunological features that correlated with CSF leukocyte counts (Table 1 and Fig 1). Among participants, 51.5% (206/400) were antiretroviral therapy (ART)-experienced, with median CD4$^+$ T cells/μL of 16 (IQR; 6–43). The CSF leukocytes were generally low (median of <5 [IQR; <5–45] cells/μL). The CSF fungal growth on the laboratory culture was at a median of 52,000 colony forming units (CFUs) (IQR; 1,195 to 335,000 fungal CFUs/mL).

The baseline participant demographic characteristics, clinical, and laboratory features (Table 1) and peripheral white blood cell count (Fig 1) did not differ by CSF leukocyte stratification (Fig 1A(i)). However, both the quantitative CSF protein (Fig 1A(ii)) and the frequency of circulating peripheral CD4$^+$ T cells and CD8$^+$ T cell count (Fig 1B(i-ii) respectively) correlated positively with the frequency of CSF leucocyte counts. In contrast, fungal burden correlated negatively (inversely) with the frequency of CSF leukocytes (Fig 1C); (Pearson r, -0.270 (95% CI; -0.358 to -0.176) and p<0.001). There was no significant statistical association between the frequency of circulating peripheral white blood cells and CSF white blood cells (Fig 1D). These findings did not differ significantly multiple co-variate comparative adjustments (S1 Table).

**Table 1. Participants Baseline Demographics and Clinical Features by CSF Leukocytes.**

| Baseline Demographics | ≤50 CSF white cells/μL | 51–200 CSF white cells/μL | 201–500 CSF white cells/μL |
|---|---|---|---|
| N | 318 | 57 | 26 |
| Age, years | 35 (29–41) | 32 (29–38) | 33 (28–39) |
| Females, n (%) | 127 (39.9) | 22 (39.6) | 11 (42.3) |
| Weight, kg | 51.5 (50–60) | 55 (50–60) | 50 (49.5–60) |
| Months on ART* | 10.0 (1.3–39.5) | 6.4 (1.2–38.6) | 2.7 (0.4–4.5) |
| CSF glucose, mg/dL# | 104.4 (88.2–122.0) | 96.0 (88.2–107.0) | 87.9 (72.5–119.7) |
| Duration of headache, days | 14 (7–30) | 14 (7.5–30) | 14 (7–23.3) |
| GCS <15, n (%) | 146 (67.3) | 29 (50.9) | 17 (65.4) |
| Hemoglobin, g/dL | 11.7 (10.1–13.3) | 11.4 (10.1–13.6) | 10.5 (9.2–12.4) |
| Platelets, x10³/μL | 192 (132–251) | 203 (145–254) | 225.5 (136–329) |
| CSF opening pressure, mmHg | 19.7 (14.7–22.1) | 17.7 (13.2–20.9) | 19.1 (15.3–20.1) |

Values are median (IQR). Statistic: Kruskal Wallis test (ANOVA) comparing the variables among participants across the three CSF white cell class intervals. Not statistically significant were variables with p-value ≥0.05 at a 95% confidence interval. ART* experience—antiretroviral therapy; reported among 165 participants with ≤50 cells/μL white cells, 32 participants with 51–200 white cells/μL, and 14 participants with 201–500 white cells/μL. CSF glucose #—reported among 145 participants with ≤50 cells/μL white cells, 24 participants with 51–200 white cells/μL, and 12 participants with 201–500 white cells/μL. GCS—Glasgow Coma Scale. CFU–colony forming units in CSF fungal growth culture. Normal adults CSF opening pressure is <10–15 mmHg that shows generally high CSF opening pressure above normal among patients with cryptococcal meningitis.

## 3.2. CSF cytokines and chemokine concentration positively correlate with the frequency of CSF leukocyte infiltration

Next, we determined among continuous variables, association of the levels of CSF cytokine and chemokine concentration in relation to the frequency of CSF leukocyte count. The level of Th1 cytokines (IL-2, IFN-γ and TNF-α) correlated with the frequency of CSF leukocyte count (Fig 2A(i-iii)). Similarly, levels of Th17 cytokine IL-17, the immune regulatory element IL-10 cytokine, and immune checkpoint PD-L1 also significantly correlated with the frequency of CSF leukocyte count (Fig 2A–2C), as did levels of the CXCR3+ T cell chemoattractant chemokine CXCL10/IP-10 and myeloid cell chemoattractant chemokine CCL11/Eotaxin (Fig 2C and 2D). These observations did not differ much with multiple co-variable comparative adjustment (S1 Table).

## 3.3. Paucity of cellular and soluble immune activated response is associated with low probability of 18-week survival

The overall 18-week survival was 55% (203/369 of participants). The survival of participants at 18-weeks was associated with elevated frequency of CSF leukocyte counts (Fig 3). The probability of survival was lowest among individuals with median frequencies of CSF leukocytes <5 cell/μL (range <5 - ≤50 cells/μL) vs. 110 cells/ μL (range 51–200 cells/μL) (51.7% vs. 67.3% survival; p = 0.032), which was comparable with a median frequency of 268 CSF leukocytes (range 201–500 cells/μL); (67.3% vs. 68% survival; p = 0.926), (Overall for 3-groups comparison, Log Rank p = 0.028) (Fig 3B). This relationship between CSF WBC count and survival was consistent whether using 5, 3 or 2 (Fig 3C) strata (Hazard Ratio, (HR = 1.634, 95% CI; 1.140 to 2.343) and p = 0.008) (Fig 3C and S3 Table).

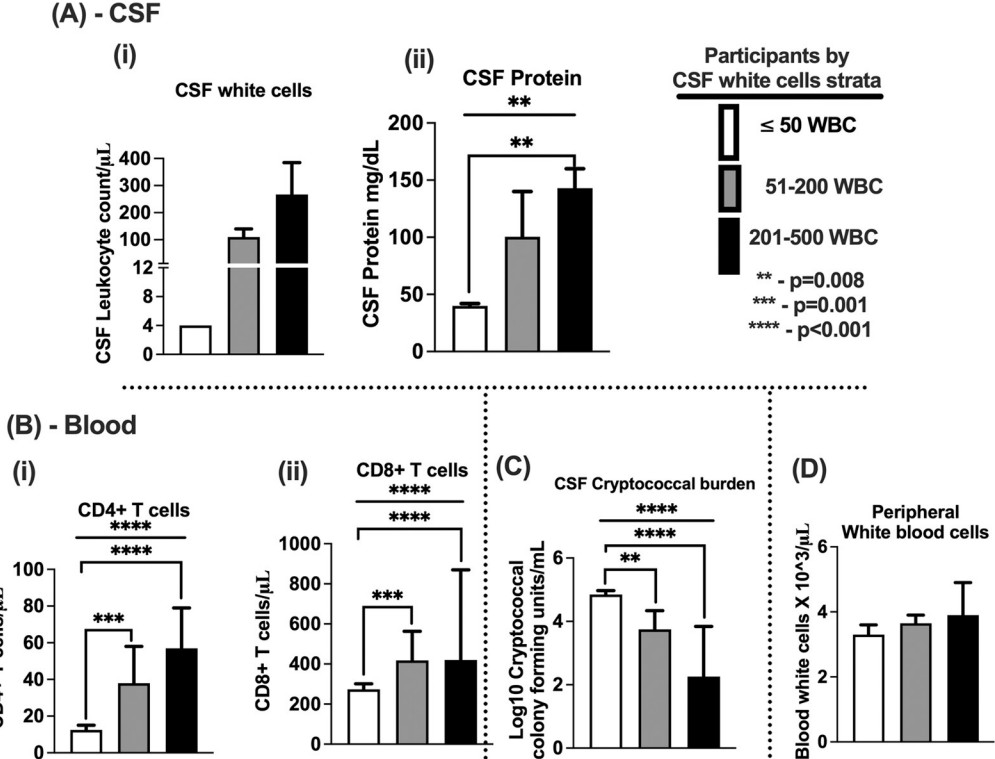

**Fig 1. Unadjusted association of CSF white blood cells with CSF and blood clinical features.** A (i)–levels of stratified CSF white blood cells. A (ii)–levels of CSF proteins. B (i)–levels of peripheral blood CD4$^+$ T cells. B (ii)–levels of peripheral blood–CD8$^+$ T cells. C–CSF fungal burden. D–peripheral white blood cells. The interlinking bars–shows two variable unpaired comparison. The flat bar shows three variables analysis of variance (ANOVA) comparisons. The error bars show median and 95% confidence intervals (CI). A-(i) both the medians and interquartile range was 4 cells/µL, (reported as <5 cells/µL). Asterisk *—show statistically significant variables reported at p-value <0.050, at 95% CI. §The flat bar shows three variables analysis using ANOVA test.

Compared to patients who died, survivors had significantly lower frequency of fungal growth (CFUs/mL) from CSF laboratory fungal cultures, (Fig 3D(i)), high frequency of CD4$^+$ T cells, CD8$^+$ T cells, and CSF white blood cell count/µL (Fig 3D(ii)). Also, survivors had high levels of CSF CXCL10, and IL-17A (Fig 3D(iii) compared with those who died (S4 Table). Other variables including CSF IFN-γ, CSF protein tended to be statistically lower with survival (S4 Table). In contrast, the concentration levels of PD-L1, TNF-α and IL-2 however, tended to be high among survivors compared to those who died (Figs 3D(iii) and S2).

### 3.4. Putative fungal burden, host immune and survival-related variance with principal component analysis

To integrate putative fungal, host immune, and survival-related determinants, we explored the opportunities to support intra-data simple random stratification or clustering to enable subsequent intra-cluster analysis using principal component analysis (PCA). With Eigen vector projections on principal components, (PC) 1 and PC2, (Fig 4A and S5 Table), the data stratified and reduced to a single plane 3-vector distribution, that enables within cluster variance analysis. With this approach 3-data clusters were observed; survival and cryptococcal growth determined fungal colony forming units (CFUs) clustering together (Fig 4A(i)), followed by the cytokines and chemokines that also clustered together (Fig 4A(ii)), and clinical measurements in blood and CSF that included, the frequency of circulating leukocytes in blood, CD4$^+$, CD8$^+$

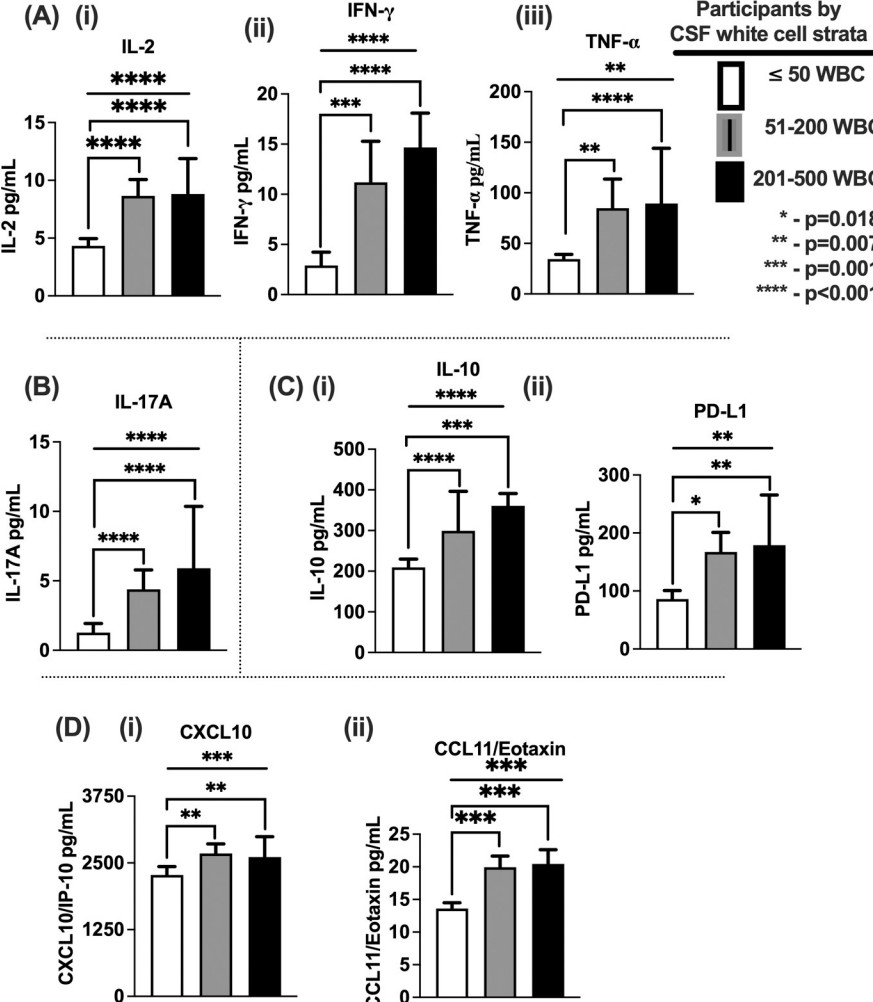

**Fig 2. Unadjusted Correlation of CSF cytokines and chemokine levels with CSF leukocyte counts.** A- Th1 cytokines; A (i)—Interleukin 2, A (ii)—Interferon gamma, A (iii)—Tumor necrosis factor alpha. B—Th17 cytokine, IL-17A. C–Immune regulatory elements; (i)–Interleukin 10 (IL-10), and programmed death 1 ligand (PD-L1). D Chemokines; D (i) CXCL10/IP-10 and D (ii)–CCL11/Eotaxin. The CSF white cells; ($\leq$50 cells/μL; n = 318), (51–200 cells/μL; n = 57) and (201–500 cells/μL; n = 26) participants. The interlinking bars–shows two variable unpaired comparison. Error bars–show median and 95% CI. The flat bar shows three variables ANOVA comparisons. Asterisks *—show statistically significant variables reported at p-value <0.050, at 95% confidence intervals.

T cell, CSF leukocyte and the levels of CSF protein concentration) (Fig 4A(iii)). Thus, within, between, and among clusters further univariate or multivariate analysis is used to determine independent factors influencing model outcome.

By principal components analysis (PCA), simple random association existed between 18-weeks survival and baseline cryptococcal Log10 CFU/mL (Fig 4A(i)). These principal components (18-weeks survival and baseline cryptococcal Log10 CFU/mL), (Fig 4A(i)), related orthogonally with CSF cytokines and chemokine responses (Fig 4A(ii)) and diagonally with clinical measurements in blood and CSF parameters (Fig 4A(iii)). The existing independent determinants or outcome predictor variables to the PCA random assigned relationships are statistically explored and discussed in the subsequent sections.

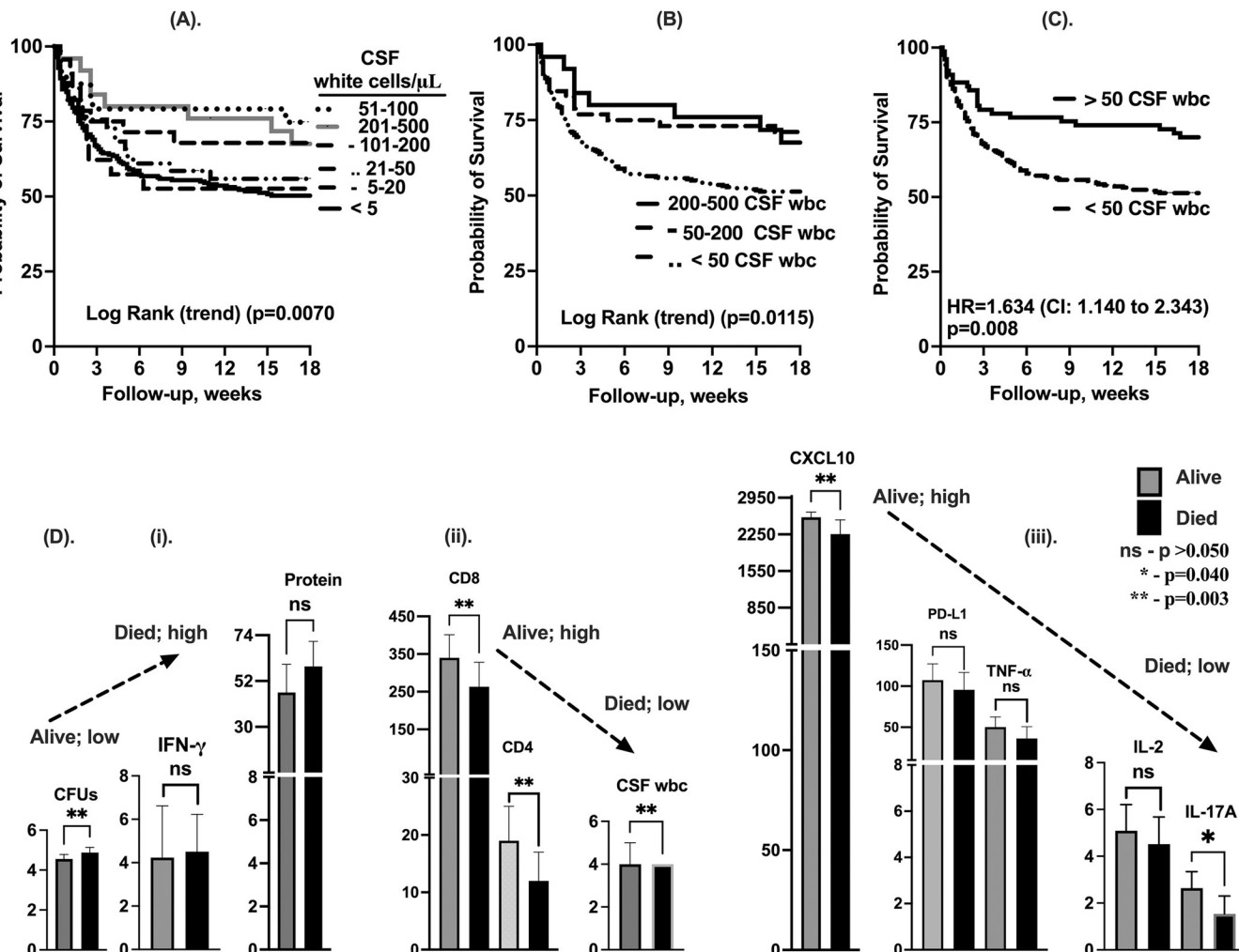

**Fig 3. Correlation of CSF white cells with 18-week survival.** A–survival by CSF white cell intervals (<5 cells/μL; n = 245), (5–20 cells/μL; n = 31), (21–50 cells/μL; n = 42), (51–100 cells/μL; n = 26), (101–200 cells/μL; n = 31), and (201–500 cells/μL; n = 26). B—18 weeks survival by CSF white cells; (≤50 cells/μL; n = 318), (51–200 cells/μL; n = 57) and (201–500 cells/μL; n = 26) participants. C– 18-week survival with CSF ≤50 cells/μL. D (i-iii)–illustrates trends in immune responses between survivors and those who died during 18-weeks of follow-up. Statistics—Mann-Whitney unpaired t-test. *—show statically significant variables. NS- not significant. Error bars–show 95% confidence intervals. p-values, p<0.050 were statistically significant.

### 3.5. CSF leukocyte influx predict variance in circulating levels of cellular and CSF soluble immune response, and CSF fungal burden with cryptococcal meningitis

The principal component with highest effect sizes are those that contribute the highest variance to the model outcome. Cumulatively, the frequency of CSF leukocytes with an effect size of 35.7% and CSF fungal burden with an effect size of 12.8%, contributed the most influence to PC1 and PC2 variance (Fig 4B and S5 Table).

While exploring the independent association and predictor variables that existed between the frequency of CSF leukocytes with the most effect to the variance (Fig 4B) and other principal components, the frequency of CSF leukocytes was inversely related and independently predicted by the CSF fungal burden $\log_{10}$ CFUs/mL (p = 0.0022) (Table 2, model 1). In cryptococcal meningitis, given the inverse association of CSF cryptococcal fungal burden with CSF leukocyte count, the higher CSF fungal burden and/or fungal titer is either inhibitory,

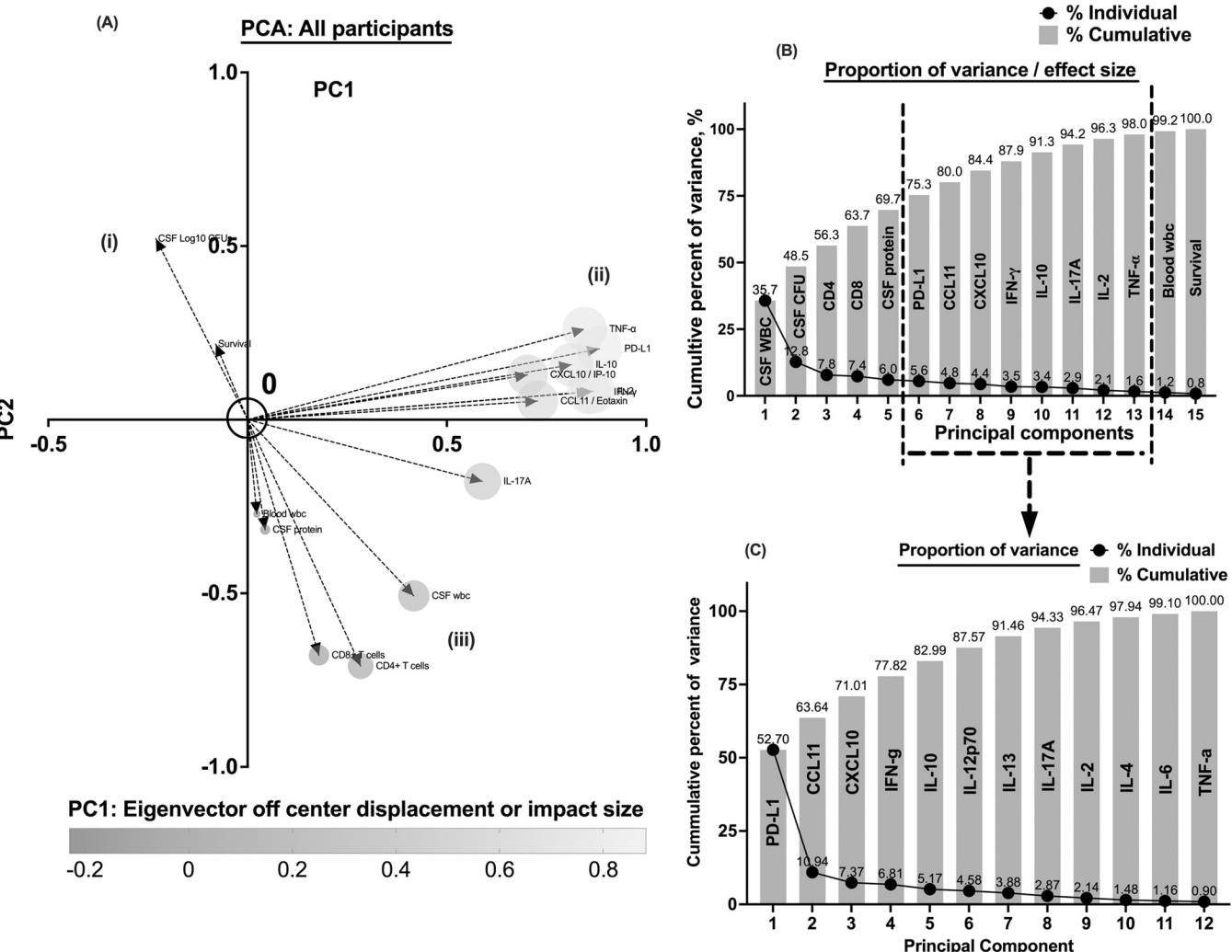

**Fig 4. Principal Component Analyses showing data clusters and variations on principal component, PC1 and PC2.** (A)—(I & ii) orthogonal Eigenvectors showing clustering and variation of the principal components between CSF fungal burden and survival with the expressed cytokine/chemokine profile. (i & iii) diagonal Eigenvectors showing clustering and variation of the principal components between CSF fungal burden and survival with CSF fungal burden and host survival and CSF white cells, CSF protein, peripheral white cells, CD4+ and CD8+ T cells. Eigenvectors projection at >5 from the center of the plane shows greater power of the principal components to predict the outcome. Also, the furthest the component to the cluster variation among all principal components. C —contribution of each principal component to the cluster variation among a subset (cytokines and chemokines) principal components.

destructive or repulsive to infiltrating CSF leukocytes and vice versa with cryptococcal meningitis.

Also, the frequency of CSF leukocytes was positively related and independently predicted by the frequency of peripheral circulating CD4+ T cells (p = 0.0119), CD8+ T cells (p = 0.0015), and CSF protein concentration (p = 0.0143) (Table 2, Model 1). Given the positive association of the circulating frequencies of CSF infiltrating leukocytes and peripheral circulating CD4+ T cells and CD8+ T cells, the circulating immunocytes is an important determinant or source of CSF infiltrating leukocytes in CSF with cryptococcal meningitis.

And, the frequency of CSF leukocytes was positively related and independently predicted by the CSF Th17 cytokine; IL-17A (p<0.0001) and Th1 cytokine; TNF-α (p = 0.0261) concentration (Table 2, Model 2). Given the positive correlation association of the CSF leukocytes and measured CSF protein and cytokine levels, the CSF leukocytes and other neuroimmune

**Table 2. Independent Immune Predictors of the Frequency of Cerebrospinal Fluid Leukocyte influx with Cryptococcal Meningitis.**

| Independent variable | Estimated regression coefficient | Standard error | 95% CI (asymptotic) | \|t\| Statistic | P value |
|---|---|---|---|---|---|
| Model 1; (n = 296): Principal component cluster (iii) CSF leukocytes dependent–independent outcome predictors determined after adjusting for CSF cryptococcal fungal burden, CD4+ T cells, CD8 T cells, CSF protein, and blood leukocytes. | | | | | |
| Model 1 Intercept | 42.5700 | 14.5600 | 13.9200 to 71.2200 | 2.9240 | 0.0037 |
| CSF cryptococcal, log10 CFU/mL | -8.1140 | 2.6270 | -13.2800 to -2.9440 | 3.0890 | 0.0022 |
| CD4+ T cells, /µL | 0.2795 | 0.1104 | 0.0622 to 0.4968 | 2.5310 | 0.0119 |
| CD8+ T cells/µL | 0.0482 | 0.0151 | 0.0186 to 0.0779 | 3.2030 | 0.0015 |
| CSF protein mg/dL | 0.0906 | 0.0367 | 0.0183 to 0.1629 | 2.4650 | 0.0143 |
| Blood leukocytes x10^3/µL | -0.2242 | 1.3350 | -2.8510 to 2.4030 | 0.1680 | 0.8667 |
| Model 2; (n = 392): Principal component cluster (ii) CSF leukocytes dependent–independent outcome predictors determined after adjusted for B7-H1 / PD-L1, CCL11 / Eotaxin, CXCL10 / IP-10, IFN-γ, IL-10, IL-17A, IL-2 and TNF-α | | | | | |
| Model 2 Intercept | 10.2300 | 9.0050 | -7.4780 to 27.9300 | 1.1360 | 0.2568 |
| IFN-gamma, pg/mL | 0.8933 | 0.5132 | -0.1157 to 1.9020 | 1.7410 | 0.0825 |
| TNF alpha, pg/mL | -0.1557 | 0.0697 | -0.2928 to -0.0186 | 2.2330 | 0.0261 |
| IL-17A, pg/mL | 3.0440 | 0.5451 | 1.9730 to 4.1160 | 5.5850 | <0.0001 |
| B7-H1 / PD-L1, pg/mL | -0.0065 | 0.0456 | -0.0962 to 0.0832 | 0.1429 | 0.8864 |
| CCL11 / Eotaxin, pg/mL | -0.5904 | 0.5665 | -1.7040 to 0.5234 | 1.0420 | 0.2980 |
| CXCL10 / IP-10, pg/mL | 0.0040 | 0.0036 | -0.0031 to 0.0110 | 1.1100 | 0.2675 |
| IL-10, pg/mL | 0.0448 | 0.0285 | -0.0112 to 0.1008 | 1.5730 | 0.1164 |
| IL-2, pg/mL | 2.0840 | 1.4650 | -0.7949 to 4.9640 | 1.4230 | 0.1554 |

Multivariate Linear Regression (Least Squares) model of immune predictors survival. n—number with complete data used in the modeling. Regression coefficient interpretation: negative regression coefficient (-)–means a unit decrease in the independent variable measurement negatively (inversely) influences the dependent outcome relationship. Positive regression coefficient (+)–means a unit increase in the independent outcome variable measurement positively correlates with the dependent outcome variable. P-values, p<0.0500 were statistically significant. CI–confidence interval.

activated cells are the likely source of immune modulating IL-17A and Th1 cytokine; TNF-α cytokine factors in CSF with cryptococcal meningitis.

Other variables including peripheral white blood cells, PD-L1, CXCL10, CCL11, IFN-γ, IL-2 and IL-10 did not independently predict the levels of CSF leukocytes count with cryptococcal meningitis (Tables 2 and S7).

### 3.6. The levels of interleukin 2 and CXCL10 independently predict survival

With survival, after adjusting for cytokine and chemokine responses, IL-2 (p = 0.0353) and CXCL10 (p = 0.0045) independently predicted survival with cryptococcal meningitis (Tables 3, model 1 and S7). But, CSF leukocytes counts, CSF cryptococcal log10 CFUs, CD4+ T cells, CD8+ T cells, CSF protein, and blood leukocytes counts, these variables did not independently predict survival with cryptococcal meningitis among all participants with (Tables 3, Model 2 and S7).

### 3.7. Programmed death Ligand-1 levels predict variance in cytokine and chemokine responses with cryptococcal meningitis

Considering the putative biological immune regulatory relevance of PD-L1, we explored to define the existing independent relationship between PD-L1 concentration with cytokines and chemokines responses. Considering intra-cluster variance among CSF cytokine and chemokine, PD-L1 with an effect size of 52.7% and CCL11 with an effect size of 16.9% contributed

**Table 3. Independent Immune Predictors of Survival with Cryptococcal Meningitis.**

| Independent variable | Estimated regression coefficient | Standard error | 95% CI (asymptotic) | \|t\| Statistic | P value |
|---|---|---|---|---|---|
| Model 1; (n = 360): Principal component cluster (ii) survival dependent–independent outcome predictors determined after adjusted for IL-2, CXCL10 / IP-10, CCL11 / Eotaxin, B7-H1 / PD-L1, IFN-γ, IL-10, IL-17A, and TNF-α. | | | | | |
| Model 2 intercept | 0.5724 | 0.0588 | 0.4568 to 0.6880 | 9.739 | <0.0001 |
| IL-2, pg/mL | 0.0207 | 0.0098 | 0.0014 to 0.0399 | 2.1130 | 0.0353 |
| CXCL10 / IP-10, pg/mL | -6.737e-005 | 2.356e-005 | -0.0001 to -2.104e-005 | 2.8600 | 0.0045 |
| B7-H1 / PD-L1, pg/mL | -6.573e-005 | 0.0003 | -0.0006 to 0.0005 | 0.2229 | 0.8237 |
| CCL11 / Eotaxin, pg/mL | -0.0023 | 0.0037 | -0.0096 to 0.0050 | 0.6233 | 0.5335 |
| IFN-gamma, pg/mL | -0.0017 | 0.0035 | -0.0086 to 0.0052 | 0.4853 | 0.6278 |
| IL-10, pg/mL | -0.0001 | 0.0002 | -0.0005 to 0.0002 | 0.7963 | 0.4264 |
| IL-17A, pg/mL | -0.0029 | 0.0035 | -0.0098 to 0.0040 | 0.8248 | 0.4100 |
| TNF alpha, pg/mL | 0.0002 | 0.0005 | -0.0007 to 0.0011 | 0.4041 | 0.6864 |
| Model 2; (n = 275): Principal component cluster (iii) survival dependent–independent outcome predictors determined after adjusted for CSF leukocytes, CSF cryptococcal fungal burden, CD4+ T cells, CD8+ T cells, CSF protein, and Blood leukocytes. | | | | | |
| Model 1 Intercept | 0.3307 | 0.1019 | 0.1301 to 0.5313 | 3.2450 | 0.0013 |
| CSF leukocytes/µL | -0.0004 | 0.0004 | -0.0011 to 0.0002 | 0.8941 | 0.3720 |
| CSF cryptococcal, log10 CFU/mL | 0.0245 | 0.0183 | -0.0115 to 0.0605 | 1.3390 | 0.1817 |
| CD4+ T cells/µL | -0.0010 | 0.0008 | -0.0025 to 0.0005 | 1.3360 | 0.1828 |
| CD8+ T cells/µL | -1.621e-005 | 0.0001 | -0.0002 to 0.0002 | 0.1558 | 0.8763 |
| CSF protein, mg/dL | 0.0002 | 0.0002 | -0.0002 to 0.0007 | 0.9532 | 0.3413 |
| Blood leukocytes, x10³/µL | 0.0121 | 0.0088 | -0.0052 to 0.0293 | 1.3770 | 0.1696 |

Statistic: Multivariate Linear Regression (Least Squares) model of immune predictors survival. n—number with complete data used in the modeling. Regression coefficient interpretation: negative regression coefficient (-)–means a unit decrease in the independent variable measurement negatively (inversely) correlates with the dependent outcome variable. Positive regression coefficient (+)–means a unit increase in the independent outcome variable measurement positively correlates with the dependent outcome variable. P-values, p<0.0500 were statistically significant. CI–confidence interval.

the highest variance in the cytokine responses (Fig 4C and S7 Table). The concentration of PD-L1 in CSF was positively related and independently predicted the concentration of CXCL10 (p = 0.0202), IFN-γ (p<0.0001), TNF-α (p<0.0001), and IL-2 (p = 0.0044), and IL-17A (p = 0.0089) in CSF (Tables 4 and S8).

## 4. Discussion

Our results show that survival with HIV-associated cryptococcal meningitis is associated with greater CSF leukocyte infiltration, lower CSF fungal burden, and higher levels of neuroimmune factors in CSF. We found that individuals with CSF leukocytes <50 cells/µL had the poorest clinical and immunological profile associated with the highest CSF fungal burden, lowest number of peripheral CD4+ and CD8+ T cells, CSF cytokines, and the lowest probability of 18-week survival compared with participants who had CSF white blood cells >50 cells/µL. Among cytokines in CSF, individuals with CSF leukocytes <50 cells/µL had the lowest level of Th1 cellular growth activating cytokine IL-2, cellular activating cytokines IFN-γ, cell death activating cytokine TNF-α, and CXCR3+ T cell chemoattractant chemokine CXCL10, and Th17 T cell-activating cytokine IL-17A. In multivariate analysis, infiltrating CSF leukocyte numbers were negatively associated with CSF fungal burden, but positively associated with CSF protein concentration and peripheral circulating CD4+ and CD8+ T cells. Moreover,

**Table 4. Programmed death ligand 1 independent predictors of CSF cytokine and chemokine responses.**

| Independent variable (n = 392) | Estimated regression coefficient | Standard error | 95% CI (asymptotic) | \|t\| Statistic | P value |
|---|---|---|---|---|---|
| Intercept | 31.8100 | 9.8260 | 12.4900 to 51.1300 | 3.2370 | 0.0013 |
| CXCL10 / IP-10, pg/mL | -0.0091 | 0.0039 | -0.0168 to -0.0014 | 2.3310 | 0.0202 |
| CCL11 / Eotaxin, pg/mL | 0.2582 | 0.6263 | -0.9731 to 1.4890 | 0.4122 | 0.6804 |
| IL-2, pg/mL | 4.5930 | 1.6030 | 1.4420 to 7.7440 | 2.8660 | 0.0044 |
| IFN-gamma, pg/mL | 4.5990 | 0.5179 | 3.5810 to 5.6170 | 8.8820 | <0.0001 |
| TNF alpha, pg/mL | 0.8331 | 0.0646 | 0.7061 to 0.9602 | 12.8900 | <0.0001 |
| IL-17A, pg/mL | -1.5710 | 0.5975 | -2.7460 to -0.3963 | 2.6290 | 0.0089 |
| IL-10, pg/mL | -0.0012 | 0.0315 | -0.0631 to 0.0607 | 0.0371 | 0.9704 |

Statistic: Multivariate Linear Regression (Least Squares) model of immune predictors survival. n—number with complete data used in the modeling. Regression coefficient interpretation: negative regression coefficient (-)–means a unit decrease in the independent variable measurement negatively (inversely) influences the dependent outcome relationship. Positive regression coefficient (+)–means a unit increase in the independent outcome variable measurement positively correlates with the dependent outcome variable. P-values, p<0.0500 were statistically significant. CI–confidence interval.

elevated concentrations of CXCL10, IL-2, and PD-L1 independently predicted a higher probability of 18-week survival.

In cryptococcal meningitis, nearly 80% of the CSF white blood cells at the time of meningitis diagnosis are T cells, the majority (about 70%) of which are CD8[+] T cells [8]. Other mononuclear cells that are central to the CSF exudate include NK cells, monocytes [8] and B cells [7]. Critical to the outcome of cryptococcal infection is the balance of an activated cellular immune response with a poly-functional evoked soluble immune profile that functions in synergy [27–29]. The primary cells infected are tissue resident macrophages that provide an intracellular niche for cryptococcal replication [27–29]. To prevent intracellular cryptococcal replication requires an activated T cell helper and cytotoxic response to activate cytotoxic killing of cryptococcal infected macrophages. Cytotoxic CD8[+] T cells and NK T cells are modulated by Th1 CD4[+] T cell stimulation [30]. It is notable that *Cryptococcus* evasion of multiple components of the innate and adaptive immune system collectively predisposes hosts to virulent and lethal cryptococcal disease with severe immune suppression [27–29]. Herein we describe the dysfunctional response that occurs in advanced HIV disease when missing appropriately functioning CD4 T cells. Yet not all dysfunctional responses are created equal, and certain compensatory immune responses are associated with lower fungal burden and improved survival. Whether the observed responses could lead to development of immune based adjunctive therapies to boost immune responses to alleviate fungal induced injury, and increase survival among persons with severe immunosuppression is a question for further research.

The inverse association of the CSF fungal burden with the levels of cellular and soluble immune response and host survival implicates the profound lack of T-cell mediated immunity overall in a population of individuals with advanced HIV disease and very low CD4 counts in the evasion of Th1- and Th17-mediated control of intracellular *Cryptococcus* replication. We showed that upregulation of IL-17A is associated with lower CSF fungal burden and improved survival. In the mouse lung model, elevated IL-17 modulates cryptococcal yeast giant and titan cell differentiation to limit fungal spread across the blood-brain barrier [31–33]. The adaptive immune clearance of *Cryptococcus* requires precise activation and recruitment of cytokine and chemokine-producing T cells that serve as the cornerstone of immune protection [6]. The evasion of fungal immune control mechanisms with down-regulated Th1 (IFN-γ, TNF-α, and IL-2) and Th17 T cell cytokines (IL-17A) and CXCR3[+] T cell chemoattractant chemokine (CXCL10) responses, allows more vigorous replication of *Cryptococcus* and overwhelming

infection. Our study suggests that mortality following cryptococcal meningitis is associated with paucity of CSF CXCR3[+] T cell activating chemokine CXCL10, cellular growth activating cytokine IL-2, and immune checkpoint regulatory element PD-L1. The down regulated expression of these neuroimmune modulatory molecules in cryptococcal disease likely impairs recruitment and subsequent maturation of adaptive T cell function in response to CNS fungal replication.

In this context, Antonia *et al., (2019)* showed that low concentrations of circulating CXCL10 were associated with enhanced glycoprotein proteolytic cleavage of the CXCL10 chemokine terminus [34], a glycoprotein proteolytic activity that was observed across the CXCR3[+] receptor activating family of chemokines. Other affected chemokines included CXCL9 and CXCL11. The proteolytic effect of CXCR3[+] receptor family of activating chemokines was observed across a range of intracellular infecting pathogens including *Cryptococcus*, that significantly impaired T cell recruitment in response to evading intracellular pathogens [34]. Consistent with our results, patients with low levels of CSF CXCL10 had a high fungal burden [12], impaired fungal clearance, and increased probability of death [35]. These observations are consistent with a T cell evasion hypothesis in which the loss of T cell protective immunity impaired immune control of intracellular cryptococcal replication [29]. The intrathecal Th1 T cell produce IFN-γ, TNF-α, IL-2 activation cytokines, and CXCR3[+] T cell chemoattractant protein CXCL10 enable diverse immune cell lineages, especially T cells, NK, and NK-T cells, and myeloid cell lineages to infiltrate the CSF [31]. The decrement in CXCR3[+] receptor family of chemoattractant chemokines impairs CSF recruitment of important intervening cells with intracellular fungal activated killing capabilities. Thus these findings suggests, defect in T cell chemokine mediated recruitment and soluble cytokines productions favors intracellular fungal replication, production of high CSF fungal burden and impaired host recovery.

Among Th-1 cytokines, induction of IFN-γ facilitates activation and recruitment of effector cellular responses needed to activate intracellular killing of phagocytosed pathogen by antigen presenting cells [32]. The IFN-γ induces the chemokine CXCL10/IP-10, a chemoattractant ligand which recruits activated lymphocytes [36,37]. The disruption of IFN-γ-encoding genes increased the susceptibility of immune deficient hosts to infections and predisposed infected hosts to rapid progression to death [33,38]. Whether mutations in IFN-γ encoding genes observed in other infections contributes to high mortality from cryptococcal infection has not been investigated [12,13]. Barber *et al.*, demonstrated the importance of IFN-γ responses in modulating immune activation among intracellular *M. tuberculosis*-infected macrophages [17]. In an experimental murine model of severe T cell immunosuppression, the absence of an IFN-γ activating response to fully activate intracellular phagosome killing of ingested mycobacteria via the production of reactive oxygen and nitrogen species failed to limit the intracellular mycobacterial burden [17].

Clinically, the low frequency of peripheral CD4[+] T cell counts despite ART treatment at CM diagnosis potentially implicates ART failure or failed immune reconstitution with unmasking and/or progressive cryptococcal disease. Initiating an optimal combination of antifungals early during cryptococcal infection can constitute effective pre-emptive therapy to improve outcomes before the onset of severe disease demonstrated with high CSF fungal burden. However, early diagnosis or staging of cryptococcal infection is a challenge where patients present late with overt disease 1–2 weeks after the onset of symptoms [13,39].

## 5. Conclusion

Patients with cryptococcal meningitis who have elevated levels of CSF soluble cytokines, chemokine and immune checkpoint elements, CSF leukocytes and circulating CD4[+] and CD8[+] T

cell responses have lower fungal burden and high probability of survival. Paucity of CSF white blood cells is associated with lower levels of soluble Th1, Th17A cytokines, CXCL10 chemokines, PD-L1 immune checkpoint response, higher fungal burden, and increased probability of death. The low levels of CSF cellular and soluble immune modulatory factors associated with high fungal burden and increased probability of death implicate the impaired CSF cellular mediated recruitment of Th1 and Th17-producing cytokine cells linked to systemic immune suppression, and downregulated CXCL10 chemokine response that was significantly associated with death. We hypothesize that among persons with HIV-related cryptococcal infection, CXCR3[+] T cells are depleted leading to impaired CSF effector white blood cell recruitment, lower intrathecal cytokine and chemokine production, uncontrolled intracellular fungal replication, serious debilitating fungal meningitis, and ensuing death, in the absence of effective ART and antifungal therapy. The levels of CSF Th1 and Th17A cytokines, CXCR3[+] activating T cell chemokine (CXCL10), and PD-L1 immune checkpoint responses are modifiable T cell factors that could be manipulated using host-directed adjunctive immune therapy to improve cryptococcal disease treatment and survival outcomes.

## Supporting information

**S1 Fig. A priori study educated model that was used to inform study set CSF while blood cell level and interrogated immune responses.** Correlation of CSF cytokines and chemokine levels with CSF leukocyte counts ($\leq$50 cells/µL and >50 cells/µL by survival. A cryptococcal CSF fungal burden–log 10 CFU–colony forming units- Interleukin 2, A (ii)—Interferon gamma, A (iii)—Tumor necrosis factor alpha. B—CXCL10/IP-10. C—CCL11/Eotaxin. D–interferon gamma. E—Th17 cytokine, IL-17A. The interlinking bars–shows two variable unpaired comparison. Error bars–show median and 95% CI. Asterisks *—show statistically significant variables reported at p-value <0.050, at 95% confidence intervals.
(TIF)

**S2 Fig. Differences in immune response $\leq$50 CSF white blood cells/µL with survivors.**
(TIF)

**S1 Table. Multiple adjusted differences in the peripheral and Cerebrospinal Fluid Clinical Variables by Set Cerebrospinal Fluid Leukocyte Count.**
(DOCX)

**S2 Table. Multiple adjusted differences in the Cerebrospinal Fluid Clinical Soluble Factors by Set Cerebrospinal Fluid Leukocyte Count.**
(DOCX)

**S3 Table. The proportion of survival and the differences in survival curves by set levels of CSF white blood cells.** 1- Survival by 5 levels of CSF white blood cells. 2- Survival by 3 levels of CSF white blood cells. 3- Survival by 2 levels of white blood cells.
(DOCX)

**S4 Table. Descriptive statistics comparing CSF cytokines, chemokines, white blood cells and CSF fungal burden among cryptococcal meningitis patients who survived and those who died by 18-weeks of follow-up.**
(XLSX)

**S5 Table. Summary of principal analysis raw values.**
(XLSX)

**S6 Table. Summary of survival imputed models for participants who had missing survival data in the clinical variables' analysis.** The 7.9% of 401 participants had missing survival data. The imputation assumption adopted in the model to replace missing survival data were; (i)–that all missing participants were alive. (ii)–that all missing participants were dead. (XLSX)

**S7 Table. Summary of survival imputed models for participants who had missing survival data in the cytokine analysis.** The 7.9% of 401 participants had missing survival data. The imputation assumption adopted in the model to replace missing survival data were; (i)–that all missing participants were alive. (ii)–that all missing participants were dead. (XLSX)

**S8 Table. Programmed death ligand 1 independent predictors of CSF cytokine and chemokine responses.** (XLSX)

## Acknowledgments

We thank study participants for their involvement in the parent study. We thank the ASTRO trial team for the clinical management of patients and data collection including Jane Francis Ndyetukira, Cynthia Ahimbisibwe, Florence Kugonza, Alisat Sadiq, Catherine Nanteza, Kiiza Tadeo Kandole, Darlisha Williams, Edward Mpoza, Apio Alison, Radha Rajasingham, Mahsa Abassi, Enock Kagimu, Morris K. Rutakingirwa, Fiona Cresswell, and John Kasibante. We thank the doctorial committee mentorship support from Makerere University, especially from Bernard S. Bagaya, Freddie Bwanga, and Henry Kajumbula. We thank institutional support from IDI from Bosco Kafufu, Andrew Kambugu. We thank the PhD. program mentorship support received from; the IDI Research Capacity Building Unit especially from Barbara D. Castelnuovo, Aidah Nanvuma, Stephen Okoboi, and the Statistics unit especially Agnes Kiragga. John Hopkins University, School of Medicine and Bloomberg School of Public Health, Department of Molecular Microbiology, and Immunology, especially from Robert R. Bollinger and Arturo Casadevall. From the University of Colorado, Denver, Anschutz Medical Campus Aurora, Division of Infectious Disease, Department of Medicine and Veteran Affairs Research Services especially from Brent Palmer, Tina Powell, and Jeremy Rakhola. We thank the statistical data analysis mentorship received from the University of Minnesota, especially from Ananta S. Bangdiwala and Kathy Huppler Hullsiek.

## Author Contributions

**Conceptualization:** Samuel Okurut, David R. Boulware, Yukari C. Manabe, Joshua Rhein, Joseph O. Olobo, Edward N. Janoff, David B. Meya.

**Data curation:** Samuel Okurut, David R. Boulware, Yukari C. Manabe, Lillian Tugume, Caleb P. Skipper, Kenneth Ssebambulidde, Joshua Rhein, Abdu K. Musubire, Andrew Akampurira, Elizabeth C. Okafor, Joseph O. Olobo, Edward N. Janoff, David B. Meya.

**Formal analysis:** Samuel Okurut, David R. Boulware, Yukari C. Manabe, Lillian Tugume, Caleb P. Skipper, Joshua Rhein, Abdu K. Musubire, Andrew Akampurira, Elizabeth C. Okafor, Joseph O. Olobo, Edward N. Janoff, David B. Meya.

**Funding acquisition:** Samuel Okurut, David R. Boulware, Yukari C. Manabe, Joshua Rhein, Edward N. Janoff, David B. Meya.

**Investigation:** Samuel Okurut, David R. Boulware, Yukari C. Manabe, Lillian Tugume, Caleb P. Skipper, Kenneth Ssebambulidde, Joshua Rhein, Abdu K. Musubire, Joseph O. Olobo, Edward N. Janoff, David B. Meya.

**Methodology:** Samuel Okurut, David R. Boulware, Yukari C. Manabe, Joshua Rhein, Joseph O. Olobo, Edward N. Janoff, David B. Meya.

**Project administration:** Samuel Okurut, David R. Boulware, Joshua Rhein, Abdu K. Musubire, Andrew Akampurira, Edward N. Janoff, David B. Meya.

**Resources:** Samuel Okurut, David R. Boulware, Yukari C. Manabe, Joshua Rhein, Edward N. Janoff, David B. Meya.

**Software:** Samuel Okurut, Yukari C. Manabe, David B. Meya.

**Supervision:** Samuel Okurut, David R. Boulware, Yukari C. Manabe, Joseph O. Olobo, Edward N. Janoff, David B. Meya.

**Validation:** Samuel Okurut, David R. Boulware, Yukari C. Manabe, Lillian Tugume, Caleb P. Skipper, Kenneth Ssebambulidde, Joshua Rhein, Abdu K. Musubire, Andrew Akampurira, Elizabeth C. Okafor, Joseph O. Olobo, Edward N. Janoff, David B. Meya.

**Visualization:** Samuel Okurut, Yukari C. Manabe, Lillian Tugume, Caleb P. Skipper, Kenneth Ssebambulidde, Joshua Rhein, Abdu K. Musubire, Andrew Akampurira, Elizabeth C. Okafor, Joseph O. Olobo, Edward N. Janoff, David B. Meya.

**Writing – original draft:** Samuel Okurut, David R. Boulware, Yukari C. Manabe, Lillian Tugume, Caleb P. Skipper, Kenneth Ssebambulidde, Joshua Rhein, Abdu K. Musubire, Andrew Akampurira, Elizabeth C. Okafor, Joseph O. Olobo, Edward N. Janoff, David B. Meya.

**Writing – review & editing:** Samuel Okurut, David R. Boulware, Yukari C. Manabe, Lillian Tugume, Caleb P. Skipper, Kenneth Ssebambulidde, Joshua Rhein, Abdu K. Musubire, Andrew Akampurira, Elizabeth C. Okafor, Joseph O. Olobo, Edward N. Janoff, David B. Meya.

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
