## [Decision Letter · Decision Letter 0]

12 Aug 2024

Dear Dr. Okurut,

Thank you very much for submitting your manuscript "Impact of Cerebrospinal Fluid Leukocyte Infiltration and Neuroimmmune Mediators on Survival with HIV-Associated Cryptococcal Meningitis" for consideration at PLOS Neglected Tropical Diseases. As with all papers reviewed by the journal, your manuscript was reviewed by members of the editorial board and by several independent reviewers. In light of the reviews (below this email), we would like to invite the resubmission of a significantly-revised version that takes into account the reviewers' comments. Please observe that the major comments flagged by Reviewer 2 must all be fully addressed in the next version please.

We cannot make any decision about publication until we have seen the revised manuscript and your response to the reviewers' comments. Your revised manuscript is also likely to be sent to reviewers for further evaluation.

Sincerely,

Nelesh Govender, FRCPath

Academic Editor

Marcio Rodrigues

Section Editor

The major methodological issues and comments flagged by Reviewer 2 must all be fully addressed in the next version please.

Reviewer's Responses to Questions

**Key Review Criteria Required for Acceptance?**

**Methods**

-Are the objectives of the study clearly articulated with a clear testable hypothesis stated?

-Is the study design appropriate to address the stated objectives?

-Is the population clearly described and appropriate for the hypothesis being tested?

-Is the sample size sufficient to ensure adequate power to address the hypothesis being tested?

-Were correct statistical analysis used to support conclusions?

-Are there concerns about ethical or regulatory requirements being met?

Reviewer #1: The work methodology is well described and follows scientific standards

Reviewer #2: Please see summary

Reviewer #3: (No Response)

**Results**

-Does the analysis presented match the analysis plan?

-Are the results clearly and completely presented?

-Are the figures (Tables, Images) of sufficient quality for clarity?

Reviewer #1: The present study expands knowledge about the immune response in the CNS in patients with cryptococcal meningitis. The authors focus on immune elements present in CSF. However, for greater completeness of their findings, they could include in their results the antigenic titer (CrAg titer), as well as the measurement of glucose in the CNS.

Reviewer #2: Please see summary

Reviewer #3: (No Response)

**Conclusions**

-Are the conclusions supported by the data presented?

-Are the limitations of analysis clearly described?

-Do the authors discuss how these data can be helpful to advance our understanding of the topic under study?

-Is public health relevance addressed?

Reviewer #1: The conclusions of the manuscript have been extensively reviewed previously. Due to the fact that cryptococcal titers interfere with the host's immune response, it would be interesting to include these measures in the present work. Furthermore, as they probably have CSF samples, they could quickly measure glucose, which is an important factor in poor prognosis in these patients.

Reviewer #2: Please see summary

Reviewer #3: (No Response)

**Editorial and Data Presentation Modifications?**

Reviewer #1: (No Response)

Reviewer #2: Please see summary

Reviewer #3: (No Response)

**Summary and General Comments**

Reviewer #1: (No Response)

Reviewer #2: The manuscript presents an analysis of selected data from the ASTRO trial exploring associations between CSF white cell count, CSF immune markers, and survival. The overall findings are as would be expected based on prior published literature, and there are some novel data on immune markers that have not previously been studies in detail. Better CSF white cell counts are associated with higher peripheral CD4 T-cell counts and more active CSF inflammation, which is in turn associated with lower fungal burden and survival. These corroborative findings are worth publishing and add to the literature. However, the current analysis is very difficult to follow and has several major methodological limitations that significantly weaken the paper. The interpretation of the findings often goes beyond what the data show. These issues are listed below under major comments, with some suggestions as to how they could be addressed.

Major comments

1. A clear description of the study population should be given. Were they all participants in the ASTRO trial? If not, how were they selected. What treatment did they get? Why is there missing outcome data if these were trial participants? How many had missing outcome data? How and why was imputation for missing outcomes performed? And why not exclude those with missing outcomes from the survival analysis?

2. How were the primary exposure and outcome variables selected? 

In terms of CSF WCC categories, from the presented graphs, it appears that the authors split the data multiple ways then selected the categories that gave the results they needed. I presume that this was not the case, and that categories were based on biological plausible ranges. This should be clearly defined and stated in the methods. With the current categorization, the groups are very unbalanced in size. Almost everyone is in the <50 group (318) compared to just 26 in the higher group. This leads to a very large loss of power with most of the data spread in terms of WCC minimized into a single <50 category. Prior analyses have used <5 or <20 as cut points. It would seem to make much more sense to do the same here from any a priori knowledge perspective and the spread of the data. The only obvious reason to select the current categories is based on conditioning on the desired outcome (mortality) as seen in Figure 3A. From a statistical point of view this is a weak methodology and prone to bias.

In terms of mortality, why was 18 week survival chosen? Cryptococcal specific mortality, rather than that due to profound immune suppression and other OIs, is likely better captured at earlier time points. Were data similar for 2 or 10 week mortality?

3. The current analysis has a problem with multiple comparisons, and no adjustment for multiplicity in the analysis. The authors do not state how many pairwise comparisons were made, but it is clearly a considerable number. A further problem with these type of datasets is that very few of the variables are independent of each other, and likely to be very closely correlated, all really being markers of the same underlying immune response. It is therefore very difficult to use lots of individual associations or multivariable analyses using regression models to draw robust statistical conclusions. The raw data show a very clear story here, and as presented, the analysis over complicates things and makes the central message less clear. 

The authors could adjust for multiple comparisons. However, this may not be the best methodology here. It may be best to just clearly present the raw data as has been done to some extent in Figures 2 and 3. A different PCA could be performed to reduce the dimensionality of all of the combined the CSF immune markers, and then see what contributes most to PC1, PC2, etc. and how these correlate with the CSF WCC and other key variables, and outcomes.

4. The current PCA is not well described and very hard to interpret. What was added into the PC analysis? How were the components derived? What were the weightings of the individual components? How did the components differ by CSF WCC category? By mortality? Furthermore, it is stated that adjustments were then made for PC scores in other analyses. As these PC scores were presumably derived from and correlated with the parameters being included in these analyses this does not make sense. Why do this and what does it mean? The PCA plots are also very hard to interpret. Survival has been included, but is also what is being predicted in subsequent analyses using the PCA. The plot showing proportion of variance would usually show this for PC1, PC2, etc, and be used to guide which components to use.

5. The descriptions of key results in the results section are difficult to follow. For example, Line 168 “CSF cytokines and chemokine concentration positively correlate with CSF leukocytes infiltration”. Presumably the association is with the CSF WCC categories that have been set? If so, state this. Or is it as a continuous variable? And what measures of correlation were used? 

6. Likewise, when describing survival associations in Line 188, what measures of association were used? Derived from which analyses? The raw data are clear here, so perhaps just present visually and describe. 

7. “Table 2 Model 1: (PC1; (ii)) CSF white cell dependent and matched independent colinear predictor variables. Adjusted for PCA cluster.” This is very hard to interpret, both in terms of methodology used, and what the results mean. In terms of analysis, wasn’t the PCA cluser derived from the predictor variables in the first place? And how can the results be meaningfully interpreted? What is a -8 fold risk with each increase in log10 CFU? In what? This applies to all other variables in the table. 

8. Table 3 – why present both linear and logistic regressions? How were these models constructed? What do these risk ratios and odds ratios mean? It is very hard to interpret the numbers.

9. Figure 1(A) shows a statistically significant difference in CSF WCC in the CSF WCC categories that have been constructed by the authors. This is an inappropriate use of statistical inference and should be removed. Throughout Figs 1 and 2 the statistical testing takes no account of multiplicity. This should either be corrected for of a different statistical approach taken – or a justification given for why no multiplicity adjustments would be needed.

10. Fig 3 – Part (D) is hard to reconcile with text. What are the units? It looks like there is no difference in CSF IFNg between those who lived and died, no difference in protein, no difference in PD1, and no difference in IL2. This is not what is reported in text or conclusions.

11. In the discussion, Line 268, it is stated that “In this regard, the immune damage response framework suggests that antifungal therapy alone without immune modulatory therapy is not sufficient to cure the infection and modulate host immune mediated injury and death.” This is not true, and in no way supported by the data presented. With modern antifungal therapy the majority of individuals with cryptococcal meningitis will be cured.

12. Other conclusions are too far reaching and not based on the data presented. For example, in Line 274 “The inverse association of the CSF fungal burden with the levels of cellular and soluble immune response and host survival implicates the selective evasion of Th1- and Th17-mediate[d] control of intracellular Cryptococcus replication.” And Line 299 “These observations are consistent with a T cell evasion hypothesis”. Neither of these are likely to be true. There is not selective evasion of Th1 and Th17 responses or T-cell immunity – there is just a profound lack of T-cell mediated immunity overall in a population of individuals with advanced HIV disease and very low CD4 counts. If you have more CD4 cells you have more CSF immune activation and better outcomes. 

13. Further lines in the discussion could be reworded for clarity. It is hard to determine the meaning of the following sentences:

Line 330 “The decrement in these factors likely conspire to limit T cell-mediated control of intracellular cryptococcal replication in macrophages with resultant serious infection and death”. 

And

Line 339 “This exempts survival improvement tapering therapy for exacerbated inflammatory response with possible persistently elevated intracranial pressure and immune reconstitution inflammatory syndrome.”

Minor comments

14. Line 66. “Monocytes, the precursors of macrophages that harbor intracellular replication machinery for Cryptococcus”. This is not really correct. Cryptococci can replicate inside monocyte lineage cells, but they don’t “harbor replication machinery”.

15. Line 75. “The immune-activated cytokine, chemokine, and checkpoint regulatory responses are important in orchestrating sequestration of Cryptococci in the CNS”. What does this mean? Is sequestration the correct term here?

16. Line 165. “The peripheral white blood cells only had a trend to increasing with CSF white cell stratification (Figure 1 D).” A trend is a specific statistical term, and does not mean a p-value of somewhere near 0.05. Say there was no significant association.

Reviewer #3: The authors reported the associations between Cerebrospinal Fluid Leukocyte Infiltration and Neuroimmmune Mediators in HIV-Associated Cryptococcal Meningitis. Although the study provide some new insights to research area, there are some severe deficiencies in the present study:

1) the chemokines/cytokines in CSF are associated with leukocyte count and fungal burden in CSF. It is recognized that fungal burden and Neuroimmmune Mediators are potentially associated with CSF open pressure, which is a determinant of mortality of patients. the present study did not reveal the relationships between Neuroimmmune Mediators and CSF open pressure. Specially, the CSF open pressure was not included in multivariate analysis, which would produce severe bias.

2) How many patients accepted VP shunt? were there all participants did not adapt management to control CSF pressure? the effect of medical controlling of CSF on mortality is not reflected in statistics.

3) the author declared that there were 51.5% (206/400) were antiretroviral therapy (ART) experienced. So, the cytokine/chemokine concentration is probably associated with IRIS. IRIS is profoundly affecting the profile of CSF Neuroimmmune Mediators. Consequently, the mixed ART experienced and ART naive patients is not convinced on statistics.

PLOS authors have the option to publish the peer review history of their article (what does this mean?). If published, this will include your full peer review and any attached files.

Reviewer #1: Yes: Delio José Mora

Reviewer #2: No

Reviewer #3: No
---

## [Decision Letter · Decision Letter 1]

10 Dec 2024

PNTD-D-24-00912R1

Impact of Cerebrospinal Fluid Leukocyte Infiltration and Activated Neuroimmune Mediators on Survival with HIV-Associated Cryptococcal Meningitis

Dear Dr. Okurut,

Thank you for submitting your manuscript to PLOS Neglected Tropical Diseases. After careful consideration of the revised manuscript, we found that two of the three reviewers were satisfied with your revisions and expressed confidence in the improvements made. However, one reviewer has raised additional concerns that require further clarification and revision to strengthen the manuscript. These comments address important aspects that are essential for ensuring the rigor and clarity of your work. We encourage you to carefully consider these points and submit a revised version of your manuscript. Once these issues have been resolved, we will proceed with further evaluation. 

Please submit your revised manuscript within 60 days Jan 09 2025 11:59PM. If you will need more time than this to complete your revisions, please reply to this message or contact the journal office at plosntds@plos.org. Please include the following items when submitting your revised manuscript:

We look forward to receiving your revised manuscript.

Kind regards,

Marcio Rodrigues

Section Editor

Shaden Kamhawi

co-Editor-in-Chief

Paul Brindley

co-Editor-in-Chief

**Journal Requirements:**

1) Tables should not be uploaded as individual files. Please remove these files and include the Tables in your manuscript file as editable, cell-based objects. For more information about how to format tables, see our guidelines:

https://journals.plos.org/plosntds/s/tables

2) Please ensure that the funders and grant numbers match between the Financial Disclosure field and the Funding Information tab in your submission form. Note that the funders must be provided in the same order in both places as well.

**Reviewers' Comments:**

Reviewer's Responses to Questions

**Key Review Criteria Required for Acceptance?**

**Methods**

-Are the objectives of the study clearly articulated with a clear testable hypothesis stated?

-Is the study design appropriate to address the stated objectives?

-Is the population clearly described and appropriate for the hypothesis being tested?

-Is the sample size sufficient to ensure adequate power to address the hypothesis being tested?

-Were correct statistical analysis used to support conclusions?

-Are there concerns about ethical or regulatory requirements being met?

Reviewer #1: (No Response)

Reviewer #2: Please see summary

Reviewer #3: (No Response)

**Results**

-Does the analysis presented match the analysis plan?

-Are the results clearly and completely presented?

-Are the figures (Tables, Images) of sufficient quality for clarity?

Reviewer #1: (No Response)

Reviewer #2: Please see summary

Reviewer #3: (No Response)

**Conclusions**

-Are the conclusions supported by the data presented?

-Are the limitations of analysis clearly described?

-Do the authors discuss how these data can be helpful to advance our understanding of the topic under study?

-Is public health relevance addressed?

Reviewer #1: (No Response)

Reviewer #2: Please see summary

Reviewer #3: (No Response)

**Editorial and Data Presentation Modifications?**

Reviewer #1: (No Response)

Reviewer #2: Please see summary

Reviewer #3: (No Response)

**Summary and General Comments**

Reviewer #1: (No Response)

Reviewer #2: This is a revised version of an earlier submission. The manuscript has been substantially modified in light of previous comments and is markedly improved. There are a few sections where either prior comments have not been adequately addressed, or the wording remains hard to follow. These are listed below:

1. Clarification was requested about selection of trial participants. It is stated in the response that they were all ASTRO participants. However, not all ASTRO participants are included here; there are 401 participants in this analysis and 460 in the trial. How were these 401 selected? Why not include all ASTRO participants? Please specify in the methods.

2. If adjustment for multiple comparisons made no difference to the outcomes, then present all of the results correctly adjusted for multiple comparisons.

3. The following wording at line 81 is perhaps not accurately reflective of the pathology of cryptococcal disease in this population: "suggest an influence of cryptococcal and/or other underlying immune suppressive factors that selectively evade T helper (Th) - mediated defense. The evasion of the T cell helper function allows unchecked intracellular fungal replication leading to high fungal burden among patients and a lower probability of recovery." I don't think it is evasion of a functional T-cell response that is the problem here - rather it is a lack of a functional T-cell response due to the fact that these individuals have very advanced HIV disease. Could this be reworded?

4. In the results, the wording at line 216 is not clear: "This association of CSF WBC number and survival was consistent with fewer deaths (Fig 3C) and more WBC strata (Fig 3C) (stratified adjusted Hazard Ratio, (HR= 1.634, 95%

218 CI; 1.140 to 2.343) and p=0.008) (Fig 3C) and supporting materials (S3 Table)." Please reword for clarity.

5. The same comment for line 223: "These findings were statistically consisted mostly for survival with the set three white blood cells categories except for high CSF fungal burden and low levels of CXCL10 that differentiated those who died with ≤50 CSF white cells/μL from survivors (S2 Fig)." Please reword for clarity.

6. The description of the PCA remains very hard to follow. Several statements are factually incorrect. For example:

"Principal component analysis shows independent principal component clustering". This is not correct - principal components are by definition NOT clustered with each other. This is how they are derived. And "The principal component with highest effect sizes are those with the highest likelihood to predict model outcome" again is not correct; principal components do not have an intrinsic effect size - they simply reflect where there is variance within a dataset. Further rewording for clarity could help with interpretation.

7. In the discussion and conclusion, a few statements are either difficult to interpret, or over complicate the presented data. For example:

"Herein we describe a potential role of a combined polyfunctional immune elevated modulated response with influence to lower fungal burden and improve survival". What is a "combined polyfunctional immune elevated modulated response"? Could this be described more simply in a way that directly reflects the findings presented? In what way is the response "modulated"?

And

"Multiple host cellular and activated CSF soluble immune mediated elements appear to regulate adaptive-innate immune crosstalk to regulate cryptococcal fungal burden and host recovery." I don't see anything in these data supporting regulation of adaptive innate crosstalk, or how this regulation is impacting host recovery. There are very clear findings to take away from rom this work without the need to speculate or overcomplicate.

8. In Figure 1 A(i) please remove statistical testing of pre-defined white cell categories as previously requested. The bars showing medians can be presented, but it is methodologically incorrect to perform statistical testing and draw inferences about the probability of there being a true difference between these groups when you have created these groups to be different. The probability that they are different is 100% as you have made them different.

9. Table 2. Although explanation has been added, these risk ratios are very hard to interpret. How can you have a negative risk ratio? A risk ratio of 1 indicates no difference. A risk ratio above 1 indicates and increased risk. A risk ratio below 1 indicates reduced risk (e.g. RR 0.5 would indicate a 50% lower risk) What does an RR of -8 mean? Are you presenting risk ratios here or something else? A coefficient from a regression model perhaps?

10. Both risk ratios and odds ratios are still presented in the next table (with the same caveats about -ve numbers for the RRs as above) - why use both? Depending on the analysis, stick to one or the other.

Reviewer #3: (No Response)

PLOS authors have the option to publish the peer review history of their article (what does this mean?). If published, this will include your full peer review and any attached files.

Reviewer #1: **Yes: **Delio José Mora

Reviewer #2: No

Reviewer #3: No

**Figure resubmission:**
---

## [Editor Report · Decision Letter 2]

27 Jan 2025

Dear Dr. Okurut,

We are pleased to inform you that your manuscript ' Impact of Cerebrospinal Fluid Leukocyte Infiltration and Activated Neuroimmune Mediators on Survival with HIV-Associated Cryptococcal Meningitis ' has been provisionally accepted for publication in PLOS Neglected Tropical Diseases.

Thank you for your hard work in revising the manuscript. Two of the three reviewers originally assigned to this manuscript recommended acceptance of your paper, while the third was less enthusiastic. Since we were unable to have this reviewer analyze the latest version of your manuscript, we conducted an editorial-level review and concluded that your study now meets the standard required for publication in PNTDs.

Best regards,

Marcio Rodrigues

Section Editor

Shaden Kamhawi

co-Editor-in-Chief

Paul Brindley

co-Editor-in-Chief

---

## [Editor Report · Acceptance letter]

30 Jan 2025

Dear Dr. Okurut,

We are delighted to inform you that your manuscript, " Impact of Cerebrospinal Fluid Leukocyte Infiltration and Activated Neuroimmune Mediators on Survival with HIV-Associated Cryptococcal Meningitis ," has been formally accepted for publication in PLOS Neglected Tropical Diseases.

Best regards,

Shaden Kamhawi

co-Editor-in-Chief

Paul Brindley

co-Editor-in-Chief
